# Mammal Reproductive Homeobox (Rhox) Genes: An Update of Their Involvement in Reproduction and Development

**DOI:** 10.3390/genes14091685

**Published:** 2023-08-25

**Authors:** Morgane Le Beulze, Cécile Daubech, Aissatu Balde-Camara, Farah Ghieh, François Vialard

**Affiliations:** 1Equipe RHuMA, UMR-BREED, UFR Simone Veil Santé, F-78180 Montigny-le-Bretonneux, France; morgane145@hotmail.fr (M.L.B.); cecile.daubech@gmail.com (C.D.); a.baldecamara@gmail.com (A.B.-C.); farah.ghieh95@hotmail.com (F.G.); 2UFR des Sciences de la Santé Simone Veil, Université de Versailles-Saint Quentin en Yvelines—Université Paris Saclay (UVSQ), INRAE, BREED, F-78350 Jouy-en-Josas, France; 3Département de Génétique, CHI de Poissy St. Germain en Laye, F-78300 Poissy, France

**Keywords:** RHOX, homeobox, male infertility, spermatogenesis, mouse model, genetic defect

## Abstract

The reproductive homeobox on the X chromosome (RHOX) genes were first identified in the mouse during the 1990s and have a crucial role in reproduction. In various transcription factors with a key regulatory role, the homeobox sequence encodes a “homeodomain” DNA-binding motif. In the mouse, there are three clusters of Rhox genes (α, β, and γ) on the X chromosome. Each cluster shows temporal and/or quantitative collinearity, which regulates the progression of the embryonic development process. Although the RHOX family is conserved in mammals, the interspecies differences in the number of RHOX genes and pseudogenes testifies to a rich evolutionary history with several relatively recent events. In the mouse, Rhox genes are mainly expressed in reproductive tissues, and several have a role in the differentiation of primordial germ cells (*Rhox1*, *Rhox6*, and *Rhox10*) and in spermatogenesis (*Rhox1*, *Rhox8*, and *Rhox13*). Despite the lack of detailed data on human RHOX, these genes appear to be involved in the formation of germ cells because they are predominantly expressed during the early (*RHOXF1*) and late (*RHOXF2/F2B*) stages of germ cell development. Furthermore, the few variants identified to date are thought to induce or predispose to impaired spermatogenesis and severe oligozoospermia or azoospermia. In the future, research on the pathophysiology of the human RHOX genes is likely to confirm the essential role of this family in the reproductive process and might help us to better understand the various causes of infertility and characterize the associated human phenotypes.

## 1. Introduction

Infertility affects up to 15% of couples [1,2], especially due to male factors in 40–50% of cases. It can be due to hormone deficiencies, environmental exposure, infections, or immune problems [3]. Genetic aetiologies are involved in 20–30% of all cases [4].

More than 100 genes are currently known to be involved in the human male infertility phenotype [5]. The associated dysfunctions can affect the hypothalamus and pituitary gland functions, reproductive organ development, Sertoli and Leydig cells, spermatogenesis, or fertilization. More specifically, non-obstructive azoospermia (NOA) and severe oligozoospermia are known to have extensive genetic heterogeneity, due to the multiple phases of spermatogenesis that can be affected. Oligozoospermia is characterized by an extremely low sperm count. Azoospermia is defined by the absence of sperm in the ejaculate in two successive samples; it can be caused by an impairment in a secretory process (in NOA) or an excretory process (obstructive azoospermia (OA)).

Spermatogenesis is a tightly regulated process that occurs via successive phases in spermatogonia, spermatocytes, and spermatids, respectively [6]. Gene expression is regulated intrinsically, interactively, and/or extrinsically. The intrinsic program determines which genes are used and when they are expressed and involves communication between somatic and germ cells. The extrinsic program influences the interactive process via hormonal regulation [7].

Hence, a large number of genes are involved in spermatogenesis regulation in precise, cell-, and stage-specific programs [7]. 2300 genes are predominantly expressed in the testes and a comprehensive understanding of their function on spermatogenesis is necessary [8].

Here, we review current knowledge on the reproductive homeobox on the X chromosome (RHOX) genes, especially in the mice, but also in humans and other species, with a focus on spermatogenesis and fertilization. Importantly, some of the RHOX genes are promising biomarkers of male fertility in mice, but also in other species such as humans, since RHOX genes, if there is more than one in an organism, are clustered on the X chromosome.

## 2. How the Rhox Genes Were Initially Identified and Named

The Rhox genes were identified in the early 1990s, especially searching in silico for the X-linked gene Pem (placenta and embryo) originally identified as expressed in a stage-specific manner in murine embryonic development. These genes are numbered according to their physical proximity to the centromere [9,10],. Most of the information on Rhox comes from studies of the mouse model, which has the largest number of Rhox genes (see below).

By 2005, a conserved amino acid sequence and intron/exon organization has been identified for 12 genes: these included the Pem gene and also the placenta specific-homeobox Psx1 and Psx2 genes. Pem, Psx1, and Psx2 were then renamed as *Rhox5*, *Rhox6*, and *Rhox9*, respectively [11,12,13]. All these genes are close to each other and form a cluster on the X chromosome (Figure 1). In view of their proximity, expression patterns, and sequence similarity, three subclusters were defined: the proximal α subcluster (*Rhox 1* to *4*), the β subcluster (*Rhox 5* to *9*), and the distal γ subcluster (*Rhox 10* to *12*). The Rhox genes display both amino acid sequence and intron/exon organization sequence identities within their homeodomain, but are also homologous with extraembryonic, spermatogenesis, homeobox Esx1; this suggests that they were derived from a common ancestral homeobox gene [10]. The homeobox gene Esx1 is expressed in the testis, placenta, lung, and brain in humans and specifically in pre- and post-meiotic germ cells of the testis in mice. Esx1 is located far from the Rhox genes on the X chromosome [14]. The human orthologues PEPP1 and PEPP2 (PEPP stands for Pem, Esx1, Psx1, and Psx2) were identified and then renamed as *RHOXF1* and *RHOXF2* [15]; the two form a cluster and are selectively expressed in the testis (Figure 2). All members of the Rhox gene complex appear to have roles in reproduction (see below) and are selectively expressed in reproductive tissues, including the testis, ovary, epididymis, and placenta [10].

In 2006, 20 additional mouse Rhox genes were identified in the region between *Rhox4* and *Rhox5*. They are also expressed in the testis and placenta and are nearly identical to *Rhox2*, *Rhox3*, or *Rhox4*. On the basis of the sequence data, it has been suggested that these Rhox genes result from a recent duplication of the *Rhox2-3-4* trimer unit. For example, the *Rhox2* genes (a to g) have a 90 to 100% sequence identity (Table 1). Furthermore, the presence of nonsynonymous and synonymous substitutions in the homeodomain region suggests that these mouse Rhox paralogs recently duplicated and are probably creating new DNA binding sites that are not yet active [16]. The duplications occurred after the divergence of the mouse and the rat and constitute the most recently formed homeobox cluster identified to date. This cluster is an actively evolving region, with diversifying *Rhox2* and 4 paralogs selection and a neutrally evolving *Rhox3*. Gene duplication and the rapid evolution of reproductive proteins are important mechanistic elements of speciation [17]. Based on comparisons of the human and chimpanzee genomes and in line with the rapid evolution of the Rhox genes seen in the mouse, it has been also reported that *RHOXF2* has evolved in a similar way in primates [18].

Importantly, RHOX genes are expressed selectively in male and female reproductive tissues in a cell type- and region-specific manner and have key roles in embryonic development and reproduction [19].

## 3. The Rhox Genes: An Evolving Family Homeobox Genes

The homeobox is a sequence present in transcription factors containing the DNA-binding motif or “homeodomain” composed of 60 amino acids [20]. Homeodomain-containing transcription factors regulate a variety of developmental and physiological events. Many homeobox genes encode transcription factors and are found in almost all eukaryotes. There are 11 gene classes and more than 100 gene families over the course of evolution. If genes descend from a single gene in the long-extinct common ancestor of Drosophila *melanogaster* and humans, they are placed in the same family. Approximately 200 homeobox genes have been identified in mammals, and one third of these are expressed in the gonads. Tandem duplications and genome duplications have expanded the number of homeobox genes and might have contributed to the evolution of developmental complexity; however, the process of homeobox gene loss must not be ignored [21].

In animals, the largest homeobox class is formed by the Antennapedia homeobox genes. These genes are involved in the determination of pattern formation along the embryo’s anterior–posterior axis. The Antennapedia class includes the well-known HOX gene cluster, which was first identified by Lewis [22] in D. melanogaster approximately 40 years ago. The HOX cluster controls the embryo’s body plan development along the anterior–posterior axis. Family studies of HOX genes have highlighted collinearity: the genes’ order on the chromosome is the same as the order of expression along the anterior–posterior body axis during embryonic development. The RHOX genes are part of the pair-rule segmentation genes (PRD)—a diverse class that also includes the Pax genes, for example. PRD genes are not generally arranged in ancient genomic clusters, even though the double homeobox (Dux), oocyte-specific homeobox (Obox), and Rhox gene clusters arose during mammalian evolution. Like the Dux genes (found on three different chromosomes in humans), Rhox genes have expanded remarkably in rodents—albeit on the X chromosome only.

As mentioned above, the mouse Rhox genes form three clusters on the X chromosome. Each cluster shows a different type of collinearity. The genes in subcluster α display both temporal and quantitative collinearity, such that the timing and level of their expression during postnatal testis development corresponds to their position within the subcluster [10]. *Rhox1* (the most proximal gene in subcluster α) is expressed first (between 7- and 12-days post-partum (dpp)) but its expression falls off rapidly. The expression of the next gene (i.e., *Rhox2*) peaks at around 12 dpp. *Rhox3* and *4* are expressed progressively between 20 and 22 dpp. The genes in subcluster β only exhibit quantitative collinearity. Lastly, subcluster γ (*Rhox10* to *12*) presents both temporal and quantitative collinearity, like subcluster α.

## 4. The Rhox Gene Cluster: Evolution and Phylogeny

Using the Ensembl database (https://www.ensembl.org/index.html accessed on 25 June 2023) and based on the assumption of an ARX gene duplication [23], one can hypothesize that the Rhox gene appeared in Boreoeutheria about 105 million years ago, similarly to Esx1 gene orthologues. Although most mammals have similar numbers of Rhox genes, two expansions are observed: a smaller expansion in primates (79.5 million years ago) and a larger expansion in rodents and, in particular, the mouse (about 66 million years ago). Given our current knowledge of mammalian genomes, the number of Rhox genes and pseudogenes varies greatly from one species to another; however, most species (apart from rodents) have zero to two Rhox genes, but as described later, duplications of these few genes occur in other species as well, considering that some Rhox or Rhox-like genes are presently missing from species where the genome builds are not as well-defined as mice or humans (Figure 3).

After the appearance of the common ancestor of the Rhox genes about 105 million years ago, a number of evolutionary events took place. The first amplification of the gene occurred about 79.5 million years ago with the appearance of primates, which have two genes (*n* = 2). The major amplification happened about 66 million years ago, thanks to the separation of glires into two clades: rodents (*n* > 5) and lagomorphs (*n* = 0). At around the same time (about 60–63 million years ago), other amplifications took place inside the carnivora and perissodactyla orders. More recently (about 7 million years ago), another amplification occurred in humans. The human genome now contains three genes and three pseudogenes.

Comparative analyses suggest that the Rhox cluster originated in the common ancestor of primates and rodents. Positive selection has been detected in some young Rhox genes, which suggests the presence of adaptive functional diversification [24]. The mouse and rat Rhox clusters are considerably larger than their human counterpart.

The last amplification in rodents occurred about 0.5–1 million years ago. The mouse Rhox gene cluster (*n* = 42 genes and pseudogenes) is considerably larger than the rat counterpart (*n* = 14) because the former has seven to eight copies of the α subcluster paralogs. There is a 95% sequence similarity between the eight duplicated genomic regions of the mouse *Rhox2*, *3*, and *4* paralogs (see Table 1 for the *Rhox2* genes).

The mouse Rhox gene cluster contains two similar genes in the β subcluster (*Rhox6* and *Rhox9*), whereas the rat Rhox gene cluster contains only one [10,25]. The fact that (1) expansion is only observed in rodents and (2) the mouse and rat Rhox genes are disposed in almost the same manner suggests that the gene expansion arrangements occurred before the mouse/rat split [23]. Indeed, Jackson et al. also predicted that this Rhox gene duplication occurred after the mouse–rat divergence and thus gave the youngest homeobox cluster yet identified [17]. This eventful history shows that the Rhox gene cluster is evolving rapidly.

In the mouse, sequence analyses have revealed significant differences in the evolutionary signatures of *Rhox2*, *3*, and *4* paralogs, and thus, differences in selection pressure [17]. *Rhox2*, *3*, and *4* are aligned in tandem and show some degree of sequence similarity. In 2006, Jackson et al. hypothesized that *Rhox2* and *Rhox4* have diverged to perform different functions because of the complete lack of overlap in positive selection between these two genes. Moreover, the lack of sequence divergence between each 40 kb duplication unit for *Rhox2*, *3*, and *4* implies that the duplication events are relatively recent [24]. *Rhox2* and *Rhox4* have diversified in different ways under selection pressure, whereas *Rhox3* is evolving neutrally.

Lastly, MacLean et al. hypothesized that the Rhox cluster is involved in the increased reproductive capacity of rodents, relative to humans [16]. Sexual conflict, sexual selection, and sperm competition are predicted to exert strong selective pressure and to drive the rapid evolution of reproductive genes, including transcription factors [26,27].

While the phylogenetic tree for the Esx gene appears similar to that of the Rhox genes, orthologous Arx genes have been identified for all vertebrates. These results are in line with the hypothesis of an initial duplication of the Arx gene.

## 5. Sequence Identity of Mammalian Paralogs

Using the Ensembl database (https://www.ensembl.org/index.html, accessed on 25 June 2023), we evaluated the nucleotide sequence identity of mammalian Rhox genes (Table 1, Table 2 and Table 3). The human genome has three RHOX genes, two of which (*RHOXF2* and *RHOXF2B*) are duplicates with identical sequences (Table 2). The human RHOX genes are most similar to the mouse *Rhox10–14* gene (*RHOXF1*) and *Rhox6*, *-8*, and *-9* genes (*RHOXF2* and *RHOXF2B*) [28]. As previously noted, due to the extremely high degree of sequencing to differentiate *RHOXF2* and *RHOXF2B* (Table 2), we were not able to distinguish between the sequences of these two genes.

As expected, there is a strong sequence identity between human and monkeys, up to 60% for all three primates, and a 97.28% sequence identity for the bonobo *RHOXF1*. *RHOXF1* is more conserved than *RHOXF2A* and *RHOXF2B*, with 37 orthologues among 200 species (Table 3). Evaluating the copy number variations in *RHOXF2* in humans and nonhuman primates revealed a parallel gene duplication/loss in multiple primate lineages [29]. Eleven nonhuman primate species have only one copy of *RHOXF2*, humans have two copies, Old World monkeys have four, and chimpanzees have six. Duplication in primates was probably mediated via non-allelic recombination. Furthermore, the gene selection process seems on-going, analyzing non-synonymous variant sites in humans [30]. Lastly, rapid evolution and copy number changes in *RHOXF2* (driven via positive selection) act on the male reproductive system.

With regard to mouse Rhox genes, we focused solely on the *Rhox2* cluster. The degree of sequence identity ranged from 87.43% (for *Rhoxf2f* vs. *Rhoxf2d*) to 100% (for *Rhoxf2a* vs. *Rhoxf2e*) (Table 1).

## 6. The Rhox Gene Family Expression

Seven RHOX genes or pseudogenes in humans and 36 Rhox genes or pseudogenes in the mouse are listed in the main public databases (https://gtexportal.org/home/, https://www.ensembl.org/index.html, https://www.omim.org/, etc. accessed on 25 June 2023). All are located on the X chromosome (Figure 1 and Figure 2). To evaluate the expression of each RHOX/Rhox gene, we compiled data from seven different mRNA RNA-Seq databases in humans, and four in the mouse, as listed in the Expression Atlas database (https://www.ebi.ac.uk/gxa/home accessed on 25 June 2023), as previously reported [31].

For each gene, the highest level of tissue mRNA expression and the mean testis ratio (the ratio between testis expression and the total expression level for all other tissues) were reported in humans (Table 4) and mice (Table 5). Protein expression (evaluated according to the Human Protein Atlas (https://www.proteinatlas.org/ accessed on 25 June 2023) and the Human Proteome Map) was only reported for humans. The data on expression (or not) in the human testis were extracted from the Human Protein Atlas.

In humans, all three RHOX genes are testis-specific on the protein level but RHOF1 mRNA is also found in the brain; details are provided below.

In the mouse, 30 of the 33 genes or pseudogenes are expressed. When considering the 13 genes reported in two or more databases, six are highly expressed in the testis and seven are exclusively expressed in the testis. Again, details are provided below.

## 7. Rhox Gene Expression and Function in the Mouse

To date, 33 Rhox genes have been identified in the mouse genome, making it the most gene-rich homeobox cluster in any species. A 40 kb region within the Rhox cluster has been duplicated eight times in tandem, resulting in eight paralogs of *Rhox2* and *Rhox3* and seven paralogs of *Rhox4* (see above) [17]. Eight of the *Rhox2* paralogs, five of the *Rhox3* paralogs, and seven of the *Rhox4* paralogs are capable of producing full-length proteins. Therefore, at least 30 of the 33 genes in the Rhox cluster are predicted to encode functional proteins; this is over twice the number of genes present in the largest homeobox cluster identified to date in any species [17].

At present, the literature data on the regulation of Rhox genes are scarce. Oda et al. showed that a large proportion of Rhox genes in the α and β subclusters are regulated via DNA methylation [32]. The researchers suggested the existence of a methylation-targeted domain in the X chromosome that comprises the 5′ end of the Rhox gene cluster and thus allows their tissue-specific expression pattern during embryonic development. However, it is known that five Rhox genes (*Rhox2*, *3*, *5*, *10*, and *11*) are androgen-dependent, and thus, are candidates for the regulation of secondary androgen-responsive genes of importance in spermatogenesis and are upregulated via testosterone [23]. Lastly, the *Rhox5* gene is regulated by many different stimuli, including differentiation signals, oncogenic signals, hormones, and DNA methylation [33].

### 7.1. Rhox1

*Rhox1* encodes a 224-amino-acid protein that interacts with various partners, including Stra8, *Rhox5*, *Rhox6*, *Rhox11*, and *Rhox9*. This gene is expressed and shut off at key time points during the first wave of spermatogenesis [10]. *Rhox1* is expressed in Sertoli cells when they are actively dividing, suggesting that *Rhox1* promotes Sertoli cell proliferation and prevents them from undergoing premature terminal differentiation. *Rhox1* might also regulate the transcription of Sertoli cell genes that encode cell surface or secreted proteins that interact with germ cells to promote their proliferation and/or inhibit their differentiation. MacLean et al. hypothesized that because *Rhox1* is one of the most highly expressed Rhox genes in the ovary, it inhibits the premature differentiation of ovarian cells.

### 7.2. Rhox2 and Rhox4

*Rhox2* and *Rhox4* are very similar and have 65% nucleotide sequence identity. *Rhox2* and its paralogs encode 16 proteins, whose average length is 191 amino acids. The *Rhox4* gene (originally named EHOX [34]) and its paralogs encode seven proteins, whose average length is 205 amino acids. *Rhox2* and *Rhox4* are expressed at similar levels in undifferentiated embryonic stem cells (ESCs) and are downregulated when the ESCs start to differentiate. The strong expression of *Rhox2* or *Rhox4b* is incompatible with an undifferentiated ESC phenotype [17]. Furthermore, it has been reported that high levels of *Rhox2* and the *Rhox4* expression override the effects of the leukemia inhibitory factor (LIF), which inhibits differentiation, and thus, might drive ESCs to differentiate [17]. Lastly, it has been reported that *Rhox4* is also expressed in trophoblast stem cells, which suggests a role in the stem cells in the developing placenta [10].

### 7.3. Rhox3

*Rhox3* and its paralogs encode 19 proteins comprising 93 to 215 amino acids. The expression of the *RHOX3* protein is first detected at P27 in the testis and is elevated at later time points (corresponding to the round-to-elongating spermatid transition) [35]. Although *Rhox3* knockdown (using short hairpin RNA (shRNA)) results in spermatogenic defects, its role in spermatogenesis in the adult mouse is uncertain because shRNA impaired the endogenous small interfering RNA (siRNA) expression [35].

### 7.4. Rhox5

*Rhox5* (originally named Pem) is initially expressed in the unfertilized mouse oocyte, then at low levels in the blastocyst, and later mainly confined to male and female primordial germ cells (PGCs) and extra-embryonic tissues at very high levels [33,36]. After birth, the *Rhox5* expression is restricted to specific somatic cells in the reproductive tract: Sertoli cells, principal cells in the epididymis, and granulosa cells in the ovary [9,37,38].

Disruption of *Rhox5* in male mice reduces the sperm count and motility, elicits greater germ cell apoptosis, and causes subfertility [10]; these observations suggest that *Rhox5* is necessary for both the formation and maturation of spermatozoa.

*Rhox5* is also a master regulator in the murine epididymis because its deletion alters the expression of most of the other Rhox genes (mostly *Rhox13*, *Rhox12*, and *Rhox1*) [39]. These observations suggest that androgen-dependent events in the epididymis are driven via a complex network of RHOX transcription factors [40].

Lastly, *Rhox5* is expressed in all the trophoblast cell layers and in the primitive endoderm-derived extraembryonic layers. However, *Rhox5*-knockout (KO) mice appear to have a normal placenta and normal embryos [37]—possibly as the result of the compensatory expression of a similar homeobox gene, such as *Rhox6* [11].

Transcription of *Rhox5* mRNA is regulated via two promoters, the distal (Pd) and the proximal (Pp), which are independently regulated [41]. While the Pp is restricted to somatic cells in the testes and epididymis, the Pd is expressed in the early embryo and somatic cells in adult female reproductive tissues and is co-expressed with the Pp in the testes [42]. Various studies have shown that the developmental and region-specific expression pattern of the *RHOX5* homeobox transcription factor in the caput epididymis (which in turn controls the expression of genes that are critical for promoting sperm motility and function) might be regulated via a GATA transcription-factor-binding site and an androgen response element in the proximal promoter [43].

Furthermore, *RHOX5* is necessary for the expression of Insulin-2 (Ins2) during the first wave of spermatogenesis when proliferating spermatogonia are abundant. More generally, *RHOX5* regulates the levels of various metabolic genes in the testis and potentially dictates local metabolism [44]. It also regulates the insulin action regulators adiponectin (Adipoq) and resistin (Rtn) and the energy metabolism transcriptional regulators peroxisome proliferator-activated receptor γ (Pparg) and the peroxisome proliferator-activated receptor γ coactivator 1-α (Ppargc1a). These observations suggest that *RHOX5* is a key transcription factor controlling cellular metabolism events that are important for spermatogenesis.

In summary, *RHOX5* appears as a central transcription factor that promotes the survival of male germ cells via its effects on cellular metabolism.

### 7.5. Rhox6

*Rhox6* was identified by screening a mouse embryonic cDNA library and was first named placenta-specific homeobox 1 (Psx1) [45]. This gene encodes a 247-amino-acid protein, the expression of which was first detected on embryonic day 8.5 in the chorionic ectoderm and the trophoblast giant cells [11]. *Rhox6* mRNA is expressed in extraembryonic tissues (mainly in the placenta and amnion) and in the PGCs of nascent bipotential gonads—suggesting an important role in early development [13,36]. *Rhox6* knockdown impairs the differentiation of PGC-like cells in culture [46].

As the *RHOX6* protein is identical to *RHOX9,* it has been proposed that they have common functions during development. Moreover, the analysis of the shRNA-*Rhox9* effects shows that *RHOX9* regulates the expression of *Rhox6*.

### 7.6. Rhox8

*Rhox8* (originally named as Tox) encodes a 320-amino-acid protein exclusively expressed in the somatic compartment of both the testis and the ovary, from the embryonic gonad period into adulthood [36]. The fact that *Rhox8* is expressed postnatally in Sertoli “nurse” cells suggests that the protein regulates the expression of somatic cell gene products with a crucial role in germ cell development [47].

Sertoli cell-specific *Rhox8* knockdown using RNA interference (RNAi) induces subfertility in the mouse, with low sperm count, poor sperm motility, a partial block in the spermatogonium-to-spermatocyte transition, and elevated apoptosis (caused by the downregulation of Sox8 and Sox9, both of which are key genes in Sertoli cell development and function). *Rhox8* knockdown also leads to the downregulation of Ins2 and the upregulation of Gdf9, which might be responsible for greater germ cell death. As mentioned above, Ins2 is a direct target of *RHOX5* and an important factor in the cell’s metabolism. *RHOX5* and *RHOX8* appear to govern the expression of large subsets of Sertoli-specific genes [47]. Recently, the KO model has been reported [48]. An increase in testis size with dilated seminiferous tubules and rete testis was observed, leading to disrupted spermatogenic cycles, spermatogenesis defects, and low fecundity. An epididymal fluid backup or dysfunction has been hypothesized to explain the results.

### 7.7. Rhox9

*Rhox9* (also known as Psx2 and germ-line–placenta–homeobox (Gpbox)) [13] is expressed in urogenital ridges in males and especially in females, but not in adult tissues. The dimorphic expression pattern of *Rhox9* during embryonic development coincides with the emergence of the morphological features that distinguish male and female gonads, such as testis cords in males and an entrance into meiosis in females. The absence of *Rhox9* mRNA in the adult testis suggests that it does not modulate the expression of genes involved in meiosis. It has been hypothesized that *Rhox9* regulates genes involved in meiotic arrest at the dictyate stage—a process that occurs in female germ cells only. Like *Rhox6*, *Rhox9* was originally identified as a homeobox gene expressed in PGCs in the nascent bipotential gonads and the placenta [12,13]. *Rhox9*-KO mice are fertile, and their male and female gonads develop normally; hence, the gene product might not be necessary for gonadal development, or compensatory phenomena might occur [49]. As *Rhox6* and *Rhox9* have a high degree of sequence identity with fairly similar tissue expression, a *Rhox6/9*-double KO mouse model is needed to likely observe a phenotype.

### 7.8. Rhox10

During development, *Rhox10* is exclusively expressed in the cytoplasm of mitotically arrested gonocytes in the fetal gonad. In the adult mouse, *Rhox10* is the only Rhox gene to show a male-specific expression in germ cells [36]. *Rhox10* is primarily expressed in premeiotic male germ cells in the adult testes (see below), and *Rhox10* levels increase between postnatal days 3 and 9—the time during which the spermatogonia expand [50]. The *Rhox10* expression gradually declines after postnatal day 10, when the number of spermatogonia falls as a result of their conversion into meiotic spermatocytes. Given that the levels of *Rhox10* mRNA in late spermatocytes, round spermatids, and elongated spermatids are very low, *RHOX10*′s role in later germ cell events cannot not be elucidated.

After birth, *RHOX10* is expressed in the cytoplasm and nucleus of the spermatogonia. At the leptotene stage, *Rhox10* can no longer be detected in the cytoplasm, and so a hypothetical switch might restrict the *RHOX10* expression to the nucleus at this meiotic stage. In late pachytene, however, low levels of *RHOX10* are observed in the cytoplasm only, which suggests the existence of a second subcellular switching mechanism. This might result from specific mitotic and meiotic signals that impinge on *Rhox10*′s transport and/or its binding to various DNA and RNA targets in the nucleus and cytoplasm, respectively, in a developmentally regulated manner [50]. *Rhox10* is the only Rhox gene whose loss causes major spermatogenic defects, including inefficient spermatogonial stem cell (SSC) generation and the progressive loss of spermatogenesis [51].

*Rhox10* regulates an intriguing array of targets, including many genes previously known to function in SSCs [52]. It induces a transcription cascade that might lead to the transition from pro-spermatogonia to spermatogonial stem cells during embryo development. *Rhox10* activates the transcription of Doublesex- and mab-3-related transcription factor 1 (Dmrt1), which in turn activates the transcription of the zinc finger and the BTB domain-containing protein (16Zbtb16) gene—a key germ cell transcription factor [52]. Furthermore, it has been reported that when hypomethylated, *Rhox10* silences the long interspersed element class 1 (LINE1) expression and transposition in the male germline [53]. LINE1 are transposable elements that comprise about 20% of the mammalian genome, on average. The activity of LINE1 must be tightly controlled; limiting LINE1 expression and translocation is fundamental for genome integrity and health. *Rhox10* binds to the promoter region of Piwil2 (encoding a key component in the Piwi-interacting RNA pathway) and appears to drive its expression. PIWIL2 is an endonuclease that produces Piwi-interacting RNA (piRNA) intermediates, which then leads to the formation of mature piRNAs. This pathway is known to suppress the Line1 expression by methylating Line1 promoters [54]. Indeed, by stimulating the PIWIL2 expression, *RHOX10* also upregulates the expression of piRNAs, which seem to drive the methylation of LINE1 promoters in germ cells. Other Rhox genes (like *Rhox3*, *6*, *8*, or *11*) also appear to repress the LINE transposition via an as-yet unknown mechanism.

### 7.9. Rhox13

*Rhox13* was firstly described in 2008 [55] and is expressed in differentiating spermatogonia and preleptotene spermatocytes. The expression disappears during meiosis, indicating that *Rhox13* is a good candidate for regulating early germ cell differentiation events and/or the transition from mitosis to meiosis. It has been suggested that *Rhox13* has an important role in the germ cell differentiation program in both sexes [56]. *Rhox13* is regulated via NANOS2, which suppresses the production of *Rhox13* in the male fetal testis by binding to *Rhox13* and maintaining it in an inactive form. *Rhox13* is expressed in spermatogonia in adults [57].

More recently, a *Rhox13* KO experiment showed that *RHOX13* had an effect during the first wave of spermatogenesis [58]. However, most of the KO mice had a mixture of normal-appearing tubules and missing adjacent layers of spermatogenic cells. Furthermore, there were many vacuolated spaces within the seminiferous epithelia of these tubules in KO mice (suggesting germ cell loss via apoptosis) and a low sperm count in young male KO mice (but not in adults). The absence of subfertility in these males shows that either *Rhox13* is not strictly necessary for this process or compensatory phenomena occurred.

### 7.10. Rhox7, Rhox11, and Rhox12

There are very few data on *Rhox7*, *Rhox11*, and *Rhox12* expressions, and these genes have rarely been studied.

## 8. Human RHOX Gene Expression and Defects

In contrast to the large body of knowledge about mice Rhox genes, the human RHOX genes have been poorly studied. It is accentuated by the fact that the RHOX cluster (like most genes involved in reproduction [26,59]) is evolving rapidly [27,29], making the information translation between rodents and humans difficult. It is already known that RHOX genes are strongly expressed in the testis and that the corresponding proteins are expressed almost exclusively in germ cells in a developmentally regulated manner [19]. *RHOXF1* is expressed in the pachytene spermatocytes and round spermatids, while *RHOXF2/F2B* is expressed in the spermatogonia and early spermatocytes. The fact that *RHOXF2/F2B* and *RHOXF1* are predominantly expressed in the early and late stages of spermatogenesis, respectively, may have functional significance. Loss-of-function RHOX variants are good candidates for the predisposition human-male infertility. Both mouse and human RHOX family members are known to be strongly regulated via DNA methylation [32,60], and RHOX hypermethylation is associated with abnormal sperm in idiopathic, infertile patients [61]. The X-linked RHOX cluster’s possible role in human fertility is intriguing.

### 8.1. mRNA Expression

*RHOXF1* is expressed in the ovary, testis, and brain during fetal development. In adults, the *RHOXF1* expression becomes ubiquitous but is predominant in the testis. In contrast, *RHOXF2* is expressed solely in the testis during both fetal and adult stages, according to the Expression Atlas and Human Protein Atlas databases.

### 8.2. Protein Expression

The protein expression of *RHOXF1* is mainly restricted to the nucleus, where it acts as a transcription factor. This expression is induced via androgens and is found especially in the testis (not only in spermatocytes but also in Sertoli cells, spermatogonia, and Leydig cells). Furthermore, *RHOXF1* is regulated via *RHOXF2/2B*, which might explain why some target genes are regulated via both of these RHOX transcription factors [62].

*RHOXF1* and *RHOXF2* differ in their subcellular localization. In the fetal testes, *RHOXF1* is predominantly expressed in the nucleus, while *RHOXF2/2B* is mostly restricted to the cytoplasm [19]. *RHOXF2/2B* relocates to the nucleus in adulthood. *RHOXF2* is exclusively found in the testis: mainly in spermatogonia and less in spermatocytes, Sertoli cells, and Leydig cells. Like *Rhox10* in the mouse, *RHOXF2* suppresses LINE1 transposition; hence, mutations in this gene might disrupt its repressive ability [53].

Some genes (such as DNAJB1, HSPA1A, HSPA6, and HSPH1) are significantly upregulated via both *RHOXF1* and *RHOXF2/2B* [62]. Moreover, several RHOX-regulated genes encode HSP70 family chaperons, and it is known that some of these genes are involved in the pathogenesis of male infertility. Greater knowledge about the regulatory pathways involving RHOX genes seems essential to better understand the impact of their deregulation.

### 8.3. RHOX Genes, Dysregulation of Spermatogenesis, and a Predisposition to Infertility

Thanks to their role as transcription factors, the RHOX genes are key players in the regulation of gene expression. It is therefore not surprising that RHOX deregulation can lead to major dysfunctions in highly regulated processes, such as gametogenesis.

With regard to *RHOXF1* dysregulation, two variants (c.515G > A, p.Arg172His and c.522C > T, p.Asp174Asp) have been described and are known to cause severe oligozoospermia in men [62]. However, both are considered to be benign because the first variant has a frequency of close to 1% in the general population, and the second has a synonymous variant. Lastly, *RHOXF1* (along with other transcription factors) is thought be involved in lubrication disorder in women, due to its strong regulation of long non-coding RNAs [63]. Indeed, the long non-coding RNAs expressed in vaginal epithelia seems to be dysregulated in patients with lubrication disorder.

Four mutations in *RHOXF2* (−73C  > G; c.202G  > A, p.Gly74Ser; c.411C > T, p.Asn137Asn and c.679G  > A, p.Gly227Arg) are known to significantly impair *RHOXF2/2B*’s ability to upregulate the expression of several of its target genes by altering the protein’s tertiary structure [62]. The c.202G > A mutation impacts the region upstream of the homeodomain—a region that is known to be a protein–protein interaction domain in other homeobox proteins—but alone is not sufficient to cause severe oligozoospermia. Instead, this mutation might predispose individuals to infertility and/or cause subfertility. In contrast, the c.679G > A variant might cause severe oligozoospermia. According to the gnomAD browser (https://gnomad.broadinstitute.org/ accessed on 25 June 2023), all these variants have a very low estimated frequency in the general population (frequency under 5 × 10^−5^), which might explain the infertility.

A number of variants causing spermatogenesis failure have been described [30]: in particular, a guanine insertion (c.381dupG) induces a premature stop codon (p.L128Afs*34) in the *RHOXF2* or *RHOXF2B* genes in patients with impaired spermatogenesis. According to the gnomAD browser, the estimated variant frequency in the general population is 0.4%; these variants are therefore candidates for a predisposition to infertility.

## 9. Conclusions

The Rhox gene family is an evolving family homeobox gene in mammals. The region is actively evolving, as shown via the interspecies differences in the number of RHOX genes and the supposedly positive selection in humans and primates. This evolving X-linked cluster might explain the great differences between rodents and other mammals in the efficiency of spermatogenesis. In the future, research on the pathophysiology of the human RHOX genes is likely to confirm the essential role of this family in the reproductive process and might help us to better understand the various causes of infertility and characterize the associated human phenotypes.

## Figures and Tables

**Figure 1 genes-14-01685-f001:**
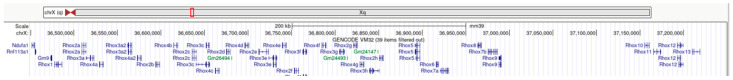
The mouse Rhox gene cluster on the X chromosome, according to the UCSC database (GRCm39/mm39): chrX:36,450,000–37,250,000.

**Figure 2 genes-14-01685-f002:**
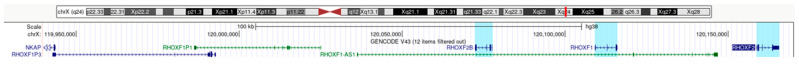
The human RHOX gene cluster on the X chromosome, according to the UCSC database (GRCh38/hg38): chrX:119,940,000–120,170,000. The genes are highlighted in blue.

**Figure 3 genes-14-01685-f003:**
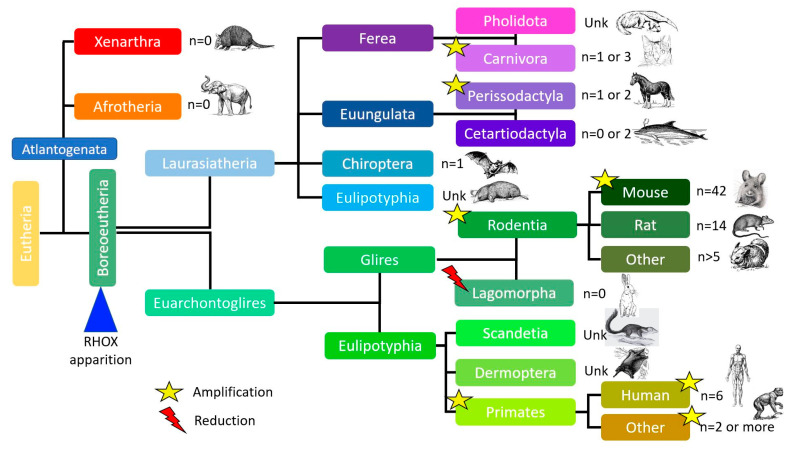
Phylogenetic tree of the Rhox gene family.

**Table 1 genes-14-01685-t001:** cDNA sequence identity of mouse *Rhox2* genes according to Ensembl database (www.ensembl.org accessed on 25 June 2023).

	*Rhox2a*	*RhoxFb*	*RhoxF2c*	*Rhox2d*	*Rhox2e*	*RhoxF2f*	*Rhox2g*	*Rhox1*
*Rhox2a*								38.22%
*RhoxFb*	97.91%							38.74%
*RhoxF2c*	97.91%	97.91%						37.70%
*Rhox2d*	90.58%	90.05%	90.58%					38.74%
*Rhox2e*	100%	97.91%	97.91%	90.58%				38.22%
*RhoxF2f*	94.76%	93.72%	93.72%	87.43%	94.76%			37.70%
*Rhox2g*	92.67%	92.15%	93.72%	92.67%	92.67%	88.48%		37.70%
*Rhox2h*	90.05%	89.53%	90.05%	90.05%	90.05%	91.10%	90.05%	38.74%

**Table 2 genes-14-01685-t002:** Nucleotide sequence identity of human RHOX genes.

	*RHOXF1*	*RHOXF2*
*RHOXF1*		
*RHOXF2*	45.12%	
*RHOXF2B*	53.26%	99.31%

**Table 3 genes-14-01685-t003:** Nucleotide sequence identity of human RHOX genes vs. Rhox genes in other mammals.

	Mouse		Rat		Specie with the Highest Sequence Identity		Orthologues Number out of 200 Species
	Gene	%Identity	Gene	%Identity	Species	%Identity	
*RHOXF1*	*Rhox13*	24.14%	*Rhox13*	23.93%	Bonobo	97.28%	37
*RHOXF2*	None	/	None	/	Gorilla	93.95%	16
*RHOXF2B*	None	/	None	/	Gorilla	93.95%	17

**Table 4 genes-14-01685-t004:** mRNA and protein expression of human RHOX genes.

RHOX Gene Number	Other Gene Name	Human													Expression in Other Vertebrates	
		Expression According to the Expression Atlas (https://www.ebi.ac.uk/gxa/home Accessed on 25 June 2023)										Location of Expression within the Testis (https://www.proteinatlas.org/ Accessed on 25 June 2023)	OMIM		Other Mammals	
		Expressed	RNA							Protein						
			Testis	Ovary	Other Tissues	Number of Experiments	Highest Tissue mRNA Level	Specificity		Highest Tissue Protein Level			Gene Number	Phenotype Number	Expression	Highest Tissue mRNA Level
								Testis Mean Ratio		Fetal	Adult					
F1P3		Yes	Low	No	Yes	3	Pancreas	0.4214	NP	NA	NA	Unknown				
F1P1		Yes	Low	Low	Yes	6	Pancreas	0.17808	NP	NA	NA	Unknown	300973	None		
F1P2		Yes	No	Low	Yes	2	Pancreas	0	NP	NA	NA	Unknown				
F1-AS1		Yes	Low	Low	Yes	8	Breast	0.053	NP	NA	NA	Unknown			Monkey, orangutan, macaque, drill, *Rattus norvegicus*, baboon, mangabey, gorilla, marmoset	
F2B		Yes	Low	No	Yes	3	Testis	1	TS	NA	Testis	Spermatogonia, pachytene and preleptotene spermatocytes			Monkey, orangutan, drill, macaque, baboon, mangabey, marmoset	
F1	OTEX, PEPP1	Yes	Medium	Low	Yes	7	Testis	0.8382	HTNS	NA	Testis	Spermatogonia and Leydig cells	300446	None	Monkey, orangutan, macaque, drill, baboon, mangabey, gorilla, marmoset, chimpanzee, hamster, mouse	Brain, testis, pituitary gland
F2	PEPP2, THG1	Yes	Medium	No	Yes	6	Testis	1	TS	NA	Testis	Spermatogonia, pachytene and preleptotene spermatocytes	300447	None	Monkey, orangutan, chimpanzee, Chinese hamster, rat, drill, macaque, gibbon, baboon, mangabey, marmoset	Testis

NA: not available; NP: no predominance; HTNS: high in testis but not specific; TS: testis-specific.

**Table 5 genes-14-01685-t005:** mRNA expression of mouse Rhox genes.

Rhox Gene	Expression According to the Expression Atlas			
	Expressed	Highest Tissue mRNA Level	Number of Experiments	Testis Mean Ratio
*Rhox1*	Yes	Testis	3	1
*Rhox2a*	Yes	Medial nasal prominence	1	
*Rhox2e*	Yes	Postnatal, day 14	1	
*Rhox3a*				
*Rhox3e*				
*Rhox4a*	Yes		1	
*Rhox2b*	Yes	Medial nasal prominence	1	
*Rhox4b*	Yes	Embryonic, day 18.5	1	
*Rhox2c*	Yes	Postnatal day 14	1	
*Rhox3c*	Yes	Testis	3	1
*Rhox4c*	Yes	Embryonic day 16.5	1	
*Rhox2d*	Yes	Testis	3	1
*Rhox4d*	Yes	Embryonic day 18.5	1	
*Rhox4e*				
*Rhox2f*	Yes	Testis	1	1
*Rhox3f*	Yes	Testis	4	1
*Rhox4f*	Yes	Maxillary arch	1	
*Rhox3g*	Yes	Mandibular arch	1	
*Rhox2g*	Yes	Medial nasal prominence	1	
*Rhox4g*	Yes	Medial nasal prominence	1	
*Rhox3h*	Yes	Testis	3	1
*Rhox2h*	Yes	Testis	1	1
*Rhox5*	Yes	Testis	4	0.9324
*Rhox6*	Yes	Kidney	3	0
*Rhox7a*	Yes	Testis	1	1
*Rhox8*	Yes	Testis	4	0.9655
*Rhox9*	Yes	Embryonic stem cell	2	
*Rhox11-ps2*	Yes	Testis	4	0.9916
*Rhox10*	Yes	Testis and skin zone	3	0.7857
*Rhox11*	Yes	Testis	4	1
*Rhox12*	Yes	Maxillary arch	1	
*Rhox13*	Yes	Testis	4	1

## Data Availability

Not applicable.

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
