# Peer review of "Mammal Reproductive Homeobox (Rhox) Genes: An Update of Their Involvement in Reproduction and Development"

_genes, 2023, doi:10.3390/genes14091685_

Round 1

Reviewer 1 Report

In the present submission, Le Beulze and colleagues provide an all-in-one review of knowledge concerning the RHOX homeobox cluster, updating previous works that focused on only expression, evolution, or function.  In general, the review is nicely written, well-referenced, flows properly, and is comprehensive in listing reports from different species.

I have some minor suggestions and clarifications that the authors can easily implement that will improve the review and have it ready for publication.  Some are very minor, but I chose to just mention them as I saw them rather than rank them by severity.

Given the number of manuscripts produced by the Vialard lab, I’m sure this is probably an artifact of the Genes submission form, but at present none of the gene names are in italics.  Although, proper care is taken to use the current MGI/JAX names and offer previous ones that were used.  It’s nice that PubMed now does that for you and shows historical papers when you search for the current name, but I like the attention to detail.

On L34, I would briefly mention what species these data come from, unless you only meant to focus on humans, but that doesn’t match with the rest of the review very well.  Perhaps add a review of rodent gene models with fertility issues and say something about the conservation of function between species (not specifically, just that it exists in many cases).

(minor) Also, probably an artifact of the submission form, but there are dozens of words where an inappropriate dash is present (i.e., L73 organi-zation).  Personally, I would forgo the full justification and not split words at the edges of paragraph boundaries (where this error probably arose).  However, if the editor does not allow that, then it will just need to be tidied up in the final draft.

For Table 1, I think the legend could be more clear about what is being compared.  Entire genomic sequence, coding sequence only, promoters included, etc.

L116 Italics for genus species

L134, remarkably, not remarkable?

L154, I think I would add right before (Figure 3) “but as described later, duplications of these few genes occur in other species as well”  I suggest this as this is something we probably don’t have a good handle on.  The present version of the mouse Rhox cluster required a good deal of manual curation because the alignment software didn’t know how to handle duplicated genes very well.  It is likely that some Rhox or Rhox-like genes are presently missing from species where the genome builds are not as well-defined as mice.  I could be entirely wrong about that, but I don’t think so.

Figure 3 is a nice addition to the Rhox field.  Out of curiosity (and maybe to add some additional meat to the review if it is known), Arx and Esx are genes that undoubtedly arose from the same precursor as the Rhox genes.  They are located outside the syntenic Rhox region in most species.  For those that you list no Rhox genes, do they also lack Esx and or Arx?  That might be of interest from an evolutionary standpoint.

L209 (and elsewhere) Minor pet peeve, feel free to ignore it if it goes against your ethos, but this is how I was trained.  Homology is ideally reserved for functional elements (i.e., two genes that have a homeodomain might be considered homologous transcription factors).  However, it’s usually not graded.  In other words, things are homologous or they are not.  The sequence “identity” describes the percentage by which two gene sequences are similar to one another.

Table 4 is almost impossible to read in printed form.  I realize remaking it from scratch and not using the screenshot is likely too annoying.  It’s possible it would be a nonissue as the online links would probably pop-out a full screen image of the table.

Table 5, I like the information, but could you describe the rationale for the order of the genes presented?  It doesn’t appear in cluster order, expression order, or paralog-grouped.  It doesn’t change the info on the table, but when I looked at it on the first pass, I thought data was missing (from prior knowledge) until I saw Rhox3c was down a bit, not in between Rhox3a and Rhox3e.

L342, there is an extra “e” in insulin

For section 7.6, a new Rhox8 knockout paper was published this week.  https://pubmed.ncbi.nlm.nih.gov/37471646/

Biol Reprod. 2023 Jul 20;ioad077. doi: 10.1093/biolre/ioad077. Online ahead of print.

Rhox8 homeobox gene ablation leads to rete testis abnormality and male subfertility in mice†

Yeongseok Oh 1 2, Maho Kasu 1, Constence J Bottoms 1, Jenna C Douglas 1, Nikola Sekulovski 2, Kanako Hayashi 1 2, James A MacLean Ii 1 2

New findings with the full gene knockout vs the Sertoli-specific knockdown you describe is that testis enlargement is present, likely due to rete testis developmental abnormality and potential epididymal fluid backup or dysfunction (the epididymis was not affected in the prior RHOX8-knockdown study as the inhibitory transgene was not expressed highly enough.

For section 7.7, I think that I would add that the Rhox9 knockout likely did not have a phenotype because Rhox6 is expressed in most tissues just as highly as Rhox9 and they have such high sequence identity.

As a side note, we looked into getting the Rhox9 knockout line and using CRISPR/CAS9 ablation to also take out Rhox6, but that line was lost.  However, we’re working on a direct generation of Rhox6/9 KO in a single step as we have done for Rhox5/Rhox8 (not published yet).

Unfortunately, my post-doc has dragged his feet on writing the DKO paper up, so I can’t give you any information to include in this review.  Between our lab and collaborators, there should be 4-5 more complete and conditional Rhox knockout papers (RNA-seq, ChiP-seq, female phenotype, etc.) that I sincerely wish were already in press to flesh out your story, heh.

For section 7.10, I think you have handled this as well as possible.  Rhox7 is a pain as its expression is variable in tissues of the same type from different animals even in littermates collected at the same time.  The commercial antibody for RHOX11 stains peritubular myoid cells beautifully.  However, I’ve seen presentations at the Society for the Study of Reproduction meetings (years ago) that show it is germ cell-specific.  Rhox12 may be in the same boat as Rhox6/9 based on published and unpublished data for expression and some sequence identity, but as you point out this has not been directly studied.

Nice review, thank you for providing this resource.

Reviewer 2 Report

The manuscript by Morgane Le Beulze et al. entitled” Mammal reproductive homeobox (Rhox) genes: an update of  their involvement in reproduction and development” is a review on Rhox genes and their function (so far) in reproduction. The review describes in details their evolution from mouse to human and function.

The present article is of general interest to the readers of the journal. The manuscript advances the knowledge in the topic. In general, the paper is well structured and well referenced.

Minor points:

In all the paper there are dashes in the middle of the words.. Example:

-line 15 differ-encis, also in line 21, 34, 73,74,81,89,93 101,106…please correct

Line 106 figure 2, I suggest to move the description above the image of the sequences.

Line 168-170: please rephrase, the concept is not clear.

Line 262: typo “.” Before table, and different character in the title. Please correct.

Line 558: in your author contribution section is missing one author role: Farah Ghieh

small make ups are needed
